# Supernova dust destruction in the magnetized turbulent ISM

Florian Kirchschlager [1,2,6], Lars Mattsson [3,6] & Frederick A. Gent [3,4,5,6]

Dust in the interstellar medium (ISM) is critical to the absorption and intensity of emission profiles used widely in astronomical observations, and necessary for star and planet formation. Supernovae (SNe) both produce and destroy ISM dust. In particular the destruction rate is difficult to assess. Theory and prior simulations of dust processing by SNe in a uniform ISM predict quite high rates of dust destruction, potentially higher than the supernova dust production rate in some cases. Here we show simulations of supernova-induced dust processing with realistic ISM dynamics including magnetic field effects and demonstrate how ISM inhomogeneity and magnetic fields inhibit dust destruction. Compared to the non-magnetic homogeneous case, the dust mass destroyed within 1 Myr per SNe is reduced by more than a factor of two, which can have a great impact on the ISM dust budget.

The interstellar medium (ISM)—gas and dust filling galactic space between the stars—is critical to galaxy evolution and, in particular, accumulating and cycling heavier elements. The survival of dust grains is a matter of debate in cosmic dust evolution studies. Several decades of research have clearly established (see refs. [1–12], among many) that a supernova (SN) shockwave provokes ion sputtering, which can efficiently destroy dust grains. The canonical model outlined by McKee[3] suggests that supernovae (SNe) can effectively cleanse dust from an ISM volume of gas mass equivalent to about 1000 M$_\odot$. Alongside sputtering of dust grains, fragmentation (as described in, e.g., refs. [6,13]) via grain–grain collisions can accelerate the destruction rate, increasing further the dust cleansing efficiency[14]. However, the net dust-destruction rate depends on complex gas dynamics, including magnetic fields and electrically charged grains.

Cosmic dust consists mainly of silicates and carbonaceous material, which profoundly impacts astronomical observations[15]. Over long timescales the atmospheres of evolved stars and molecular clouds provide the dominant channels of dust production. SNe are recognized both as intermittent producers and destroyers of cosmic dust. While ample observational evidence (e.g., refs. [16–20]) suggests a high degree of dust condensation occurs in SN remnants, there is no clear consensus on how much dust mass a typical SN shockwave may destroy due to ion sputtering or grain-grain collisions. Strong quantitative constraints on the efficiency of dust destruction are, however, of fundamental importance to correctly model the matter cycle[12,21,22]. Furthermore, dust abundances in some high-redshift galaxies appear to exceed expectations, given the current understanding of metallicity constraints and expected rates of dust destruction (see, e.g., refs. [23–25]).

Of several decisive factors determining the dust survival rate, two are essential. First, the ambient ISM gas density determines the reach of an SN blast wave and the size of the affected portion of the ISM[3]. Second, the level of shock heating and accumulation of gas in the remnant shell is critical to the sputtering and grain-grain collision rates[9,10,26,27]. Also, frictional forces between gas and dust, and Lorentz forces on charged grains determine how well gas and dust are coupled, which in turn may determine grain survival.

Treating gas and dust as dynamically separate fluids in an inhomogeneous, magnetized ISM may be of fundamental importance to the study of SN-induced dust destruction. Such a study, including all relevant forces and processes, has never been done. In particular, the effect of magnetic turbulence on the dynamics and survival of charged dust is poorly understood. In[14], shattering due to grain-grain collisions was included to model dust processing in a uniform or modestly

[1]Physics and Astronomy, Ghent University, Krijgslaan 281-S9, Ghent 9000, Belgium. [2]Physics and Astronomy, University College London, Gower Street, London WC1E 6BT, UK. [3]Nordita, KTH Royal Institute of Technology and Stockholm University, Hannes Alfvéns väg 12, Stockholm SE-106, Sweden. [4]Astroinformatics, Computer Science, Aalto University, PO Box 15399, Espoo FI-00076, Finland. [5]School of Mathematics, Statistics and Physics, Newcastle University, Newcastle NE1 7RU, UK. [6]These authors contributed equally: Florian Kirchschlager, Lars Mattsson, Frederick A. Gent. ✉e-mail: lars.mattsson@su.se

perturbed ISM, increasing the dust destruction efficiency significantly. This exacerbates the disparity between theoretical estimates of dust destruction and observed dust abundances in many starburst galaxies[28–30].

Here we apply the same dust processing models to a turbulent magnetohydrodynamic (MHD) multi-phase ISM simulation to investigate how this affects the conservation of dust abundances. Turbulence causes the dust to decouple from the gas, most so for the larger grains. Turbulence reduces the dust losses from an SN blast wave by around 10% and when the Lorentz force acting on the dust is included this increases up to 50%. The decoupling of the dust is impeded by the Lorentz force, which is important to the dust survival.

## Results

We continue an MHD simulation of a supernova-driven turbulent ISM, in which the small-scale dynamo has saturated to provide a realistic turbulent magnetic field within a turbulent multi-phase ISM. In three scenarios we explode a single remnant in a diffuse region, a region of moderate gas density and a case in which there is no explosion. We then take the same 2D slice of each case and apply two models of dust processing for a duration of 1 Myr, with and without including the effect of the Lorentz force on the dust evolution. We compare the dust destruction between all models and with a case without magnetic fields or turbulence obtained from ref. 14. The models, their labelling convention and some indicative results are listed in Table 1.

### Evolution of the MHD model

Figure 1a displays slices depicting the evolution of the gas density from a simulation of SN-driven turbulence, containing the remnant of an explosion located in low ambient density (NL or BL) and Fig. 2a an explosion located in moderate density ambient ISM (NM or BM). The models are denoted by the prefix B with the Lorentz force included and N without. The remnant retains a signature of its self-similar spherical origins at 10 kyr, but subsequent expansion is irregular due to the multi-phase ISM structure. Prior to explosion slices for all models are identical, only shifted vertically to locate suitable ambient density at each explosion epicentre. Models omitting the SN explosion are evolved separately (models NO or BO) to measure the background processing of dust, processing which occurs due to the background turbulence where the SN blast wave has not reached. Their profiles barely alter from those already visible outside the initial remnant regions at 10 kyr in Figs. 1 and 2 over 1 Myr. Maps of additional timesteps for the gas impacted by the blast waves are presented in Supplementary Figs. 1 and 2.

Corresponding temperatures are illustrated in Figs. 1b and 2b. High radiative losses from cooling in the dense medium (Fig. 2) reduce the strength of the blast wave early on. In this model reflected shock waves are weaker and damped more rapidly by its relatively higher density remnant interior. The blast wave in the diffuse gas (Fig. 1) is initially much faster and shows stronger reflected shock waves as the blast wave later encounters dense regions and propagates at high velocity in the relatively diffuse remnant interior.

Figures 1c and 2c display magnetic field strength, which has been amplified by the dynamo action of SN-driven turbulence from a subnano Gauss random seed field. By 1 Myr the magnetic field grows substantially inside the diffuse remnant, evidence of turbulent dynamo in the hot gas (see ref. 31), while in the dense remnant magnetic field is mostly evacuated with the blast wave or dissipated. The varying location and topology of the magnetic field during the evolution of each remnant may impact the dust processing.

Model NO or BO without new SNe continues to evolve the turbulence relatively slowly, the density and magnetic field profiles remaining similar to their depiction at 10 kyr in Fig. 2a and c. Velocities in the hot gas are of order 100 km s⁻¹ and in warm gas 10 km s⁻¹. Ambient ISM background dust processing is, therefore, likely to be

### Table 1 | List of model parameters and cumulative dust losses

| Model | $n_{gas,0}$ [cm⁻³] | Lorentz | SN event | 200 kyr [M⊙] | 500 kyr [M⊙] | 1 Myr [M⊙] |
|---|---|---|---|---|---|---|
| NO | ... | No | No | 1.18 | 6.36 | 18.2 |
| BO | ... | Yes | No | 0.26 | 2.42 | 8.14 |
| NL | 0.03 | No | Yes | 11.1 | 29.0 | 57.3 |
| BL | 0.03 | Yes | Yes | 5.3 | 13.8 | 28.4 |
| NM | 0.7 | No | Yes | 30.0 | 46.8 | 64.9 |
| BM | 0.7 | Yes | Yes | 18.4 | 26.5 | 37.0 |
| 14 | 1.0 | No | Yes | 48.6 | 65.0 | 70.9 |

The models to which dust processing is applied are listed and denoted by the prefix B with the Lorentz force included and N without. The MHD simulation runs are denoted by the suffix O without an SN explosion, L for an explosion in the low-density region or M for an explosion in a moderate-density region. $n_{gas,0}$ indicates the typical gas number density of the explosion epicentre at t = 0. Lorentz indicates whether or not the magnetic effects are included for the dust processing. The accumulated dust losses at 200, 500 kyr and 1 Myr are listed for each model. The ISM model of Kirchschlager et al.[14] is uniform.

significant and complex, in contrast to zero processing in a uniform ISM[10,14,27,32]. Some background processing might be discounted from the total dust losses in the SNe models.

### Dust destruction effect of magnetic fields

In Fig. 3 we illustrate the evolution of dust of different grain sizes for model BM (see Table 1). Panels a–d depict dust in bins of increasing grain size. The smallest grain size (a; 0.6 nm) is not included in the initial distribution but results from shocks and turbulence fragmenting and sputtering larger grains. Beyond 200 kyr, larger dust grains in the remnant interior are lost or advected to the shell at a rate increasing with grain size. For the same explosion where the Lorentz force is neglected in processing the dust (model NM), the annihilation of dust in the remnant interior is far more efficient. Fig. 3e shows this comparison relative to Fig. 3d for the 180 nm bin grain size. Dust destruction is significantly reduced by the presence of a magnetic field. The dust maps of the SN explosions in the low and moderate-density regions are presented in Supplementary Figs. 3 and 4.

Due to ambient turbulence, the dust processing continues outside the remnant. Such a region is identified by a white 'X' in Fig. 3. At 10 kyr the dust in these regions has a smooth distribution, scaling directly with the gas. By 200 kyr the dust decouples from the gas to form a filamentary structure. Fourth and fifth-row comparison reveals that the Lorentz force reduces the filamentary scales and contrasts in the dust. The gas velocity is the same in both models, indicating that the Lorentz force on the dust inhibits its decoupling from the gas. Inside the remnant, the large grains survive better with the Lorentz force than without.

Figure 4 shows for sample grain sizes the Pearson correlation coefficient R for the correlation between the logarithmic gas and dust number densities. Initially R = 1, but the correlation decreases over time. Dust–gas coupling is greater when the Lorentz force is present, more so the smaller the grain size (see also refs. 33, 34). Consequently, the weakest coupling occurs for the largest grains without Lorentz force after 1 Myr (see inset scatter plots in Fig. 4 and Supplementary Figs. 5 and 6).

In Fig. 5 we display the cumulative mass of dust destroyed in each model over the first Myr. For all three cases, more dust is destroyed when the Lorentz force is excluded. Background processing only (green) is negligible up to 100 kyr, after which differences due to the Lorentz force become significant, such that within 1 Myr dust losses are reduced by more than half. The rate of dust destruction increases until 200–300 kyr, becoming steady thereafter with rates of approximately 23 and 11 M⊙ Myr⁻¹ for NO and BO, respectively. This suggests an approximate 300 kyr transient redistribution of the dust from its initial

## Explosion in low density region (BL)

**Fig. 1 | Gas structure following explosion in low-density region (*BL/NL*).**
**a** Snapshots of the gas density at times $t = 10$, 200 kyr, 1 Myr. **b** Same snapshots of gas temperature. **c** Same snapshots of magnetic field strength. The SN explodes in a low-density region located at the centre of each image. A short movie showing the evolution is presented https://youtu.be/Mz2cuEVm_eY (movie frame rates $f = 133.3$ frames s$^{-1}$), from the time the SN explosion is introduced ($t = 0$) until 1 Myr later. A short movie of the turbulent ISM without SN explosion (model *BO/NO*) is available https://youtu.be/HEJ-4CZl-ZQ ($f = 33.3$ frames s$^{-1}$). The movies and data are publicly available at https://etsin.fairdata.fi/dataset/602bb9a6-0626-43db-9073-054bd3332fff/data.

gas-coupled condition into a statistically steady state, with dust depletion settling at 23 or 11 M$_\odot$ Myr$^{-1}$, respectively. Besides the total mass of destroyed dust, the dust destruction fraction of the entire domain characterizes the dust processing (see Supplementary Fig. 8) For all models presented in Table 1, the fractions are <3%.

Where explosions are sited in low (red) or moderate (purple) density gas the dust destruction is initially not very sensitive to Lorentz force effects. Magnetic effects significantly reduce dust destruction after 20–50 kyr.

### Effect of SN ambient ISM gas density on dust destruction
The first 50 kyr are critical to the effect of ambient gas density at the SN epicentre on total dust destruction. <2 M$_\odot$ of dust has been destroyed in models *NL* and *BL* (Fig. 5; red), compared to over 10 M$_\odot$ and 15 M$_\odot$ in models *NM* and *BM* (purple), respectively, or 28 M$_\odot$ for ref. 14.

Subsequently, the dust destruction rates slow considerably up to about 300 kyr. For moderate ambient density, it continues to slow, while otherwise, it increases slightly. After 400 kyr the rate of destruction becomes steady up to 1 Myr: around 57 M$_\odot$ Myr$^{-1}$ for *NL* (light-red); 30 M$_\odot$ Myr$^{-1}$ for *BL* (dark-red); 38 M$_\odot$ Myr$^{-1}$ for *NM* (light-purple); and 22 M$_\odot$ Myr$^{-1}$ for *BM* (dark-red).

The dust destruction rate is highest soon after the SN explosion, with losses higher where the ISM is dense. After about 500 kyr, the blast wave from a dense epicentre loses speed and dust losses slow as it

sweeps through more diffuse regions. From a diffuse epicentre the rate slightly increases later as more dense regions are affected.

In all cases the dust destroyed is lower than in a uniform ambient gas density of 1 cm$^{-3}$ without Lorentz force[14] (grey-dashed line in Fig. 5). The dust destruction rate is higher in ref. 14 than in other models until around 500 kyr, after which it becomes negligible.

### Impact of dust processing on dust distributions
In Fig. 6 the initial, intermediate, and final dust number densities for the models *NM* and *BM* are shown in panels b–d for selected regions identified in panel a. The dust density distributions by grain size $a$ are the averages for each box, 5 pc × 5 pc square, with insets showing initial MRN[35] (black solid line), 200 kyr (dashed lines) and 1 Myr (solid lines) profiles. Initial dust densities, following a fixed power-law spanning 5–250 nm, scale with local gas density and differ between regions.

The region at the explosion epicentre represents a more dense ISM impacted very early by the blast wave. Therefore, the dust densities (Fig. 6c) show significant alteration. After 1 Myr, the dust densities for all pre-existing grain sizes reduce by about two orders of magnitude—either swept or destroyed by the blast wave (through sputtering, fragmentation, or vaporization). Destruction is higher for large grains, especially when the Lorentz force is neglected (*NM*). With the Lorentz force, better dust–gas coupling reduces drag forces, and relative motions between gas and dust, and between grains of differing

## Explosion in moderate density region (BM)

**Fig. 2 | Gas structure following explosion in moderate density region (BM/NM).** **a** Snapshots of the gas density at times $t = 10, 200$ kyr, 1 Myr. **b** Same snapshots of gas temperature. **c** Same snapshots of magnetic field strength. The SN explodes in a moderate-density region located at the centre of each image. A short movie showing the evolution is presented https://youtu.be/oGXh6piIqlA (movie frame rates $f = 133.3$ frames s$^{-1}$), from the time the SN explosion is introduced ($t = 0$) until 1 Myr later. The movies and data are publicly available at https://etsin.fairdata.fi/dataset/602bb9a6-0626-43db-9073-054bd3332fff/data.

size. On the other hand, the Lorentz force enables a small but significant proportion of dust to get behind the shock wave, including large grains (Fig. 3d and e). Most of the destruction in this central region occurs already within 200 kyr.

Grain shattering produces a power-law distribution of smaller fragments, including a fragmental range below the short end of the initial size distribution. Destruction of pre-existing grains is lower when the Lorentz force is considered, so the final number densities at all fragment sizes are lower. Larger fragments exceeding 5 nm add to the number of densities at the short end of the initial size distribution. The outcome is a distribution well approximated by two power-laws: fragments and sputtered grains comprising the fragmental distribution and the modified initial grain size distribution with radii above 5 nm.

While fragmentation changes the shape of the grain-size density distribution, the actual dust mass destroyed by shattering is rather small. However, vaporization and sputtering can destroy dust mass, the latter especially effective for small dust grains. Fragmentation and sputtering are thus synergistic and work together by first shattering larger grains into smaller pieces and subsequently destroying dust mass by sputtering of the fragments[36].

The two other regions in Fig. 6a are located in unshocked gas beyond the 1 Myr reach of the blast wave. Alterations in the dust density distribution are caused by transport and destruction due to gas dynamics only. The unshocked region with moderate gas density

(Fig. 6d) shows a reduction of dust densities but at a significantly lower rate than in Fig. 6c. The reduction in dust densities is again larger for the model *NM* and can be assigned to dust destruction or the flow of gas from one cell to another. This gas flow is particularly crucial for the lowest density region (Fig. 6b), where the amount of dust is even increased compared to the initial conditions. The increase of gas density is 21% over the Myr. For both unshocked regions little change to the initial dust distribution occurs within the first 200 kyr (dashed lines in Fig. 6b, d).

In the lower box without the Lorentz force, higher numbers of fragmental grains within 200 kyr result from faster fragmentation of larger grains, reducing by 1 Myr to the same levels applying with Lorentz force. Given fragmentation without Lorentz force is higher at the beginning, small grain losses due to sputtering must reduce when the Lorentz force is considered.

In contrast to the other boxes, the number densities of the fragmental distribution in the upper box continue growing after 200 kyr. Much of this follows gas inflows, with which small grains are even better coupled. However, the ratio of fragmental densities to the 5 nm grain number density is smaller than in the other boxes, more so when the Lorentz force is considered, which must mainly be explained by the low fragmentation of larger grains rather than the high sputtering of small grains. As two boxes are unshocked, the alteration of dust densities indicates the importance of ISM inhomogeneity and background turbulence.

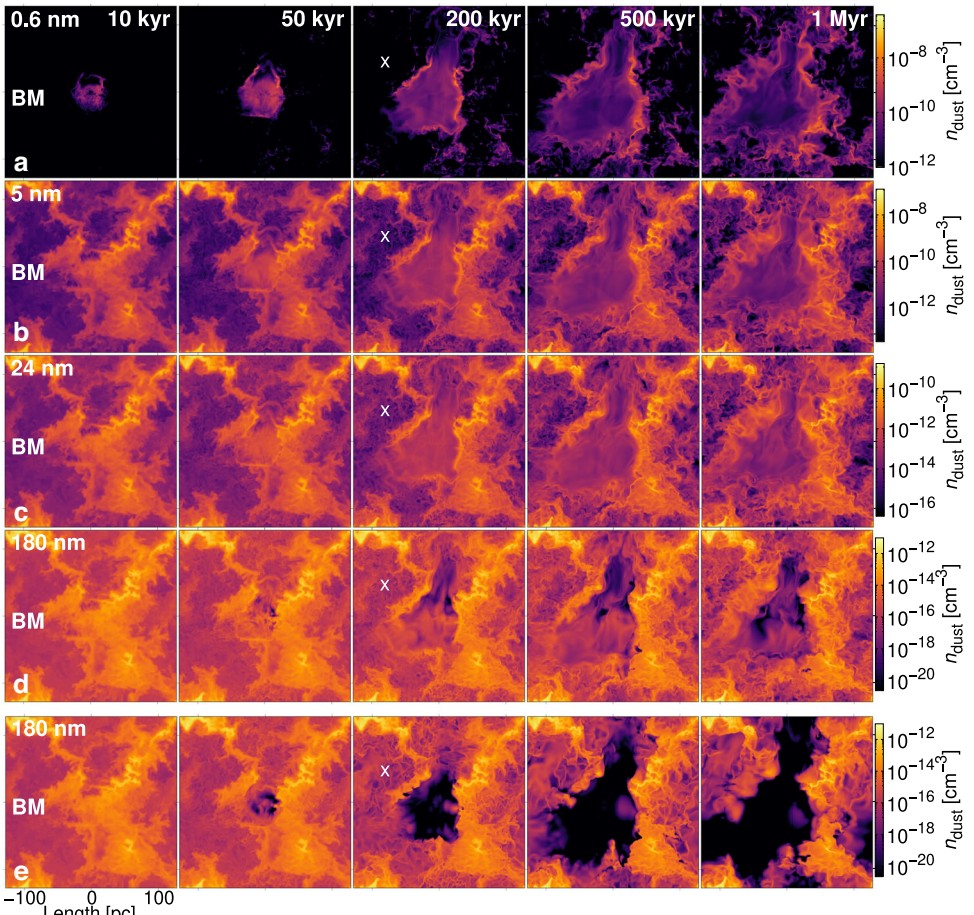

**Fig. 3 | Dust distribution for various grain sizes.** Snapshots of dust density following the SN explosion in the moderate-density region. **a**–**d** The distribution of 0.6, 5, 24 and 180 nm grains, respectively for model *BM*, which includes the effect of Lorentz force on the dust; the colour scale is fixed for each row. **e** Distribution of 180 nm grains for model *NM*, without Lorentz force effects. The white `X' identifies a region at which the Lorentz force reduces the turbulent scale of the dust (see text for details). A short movie showing the temporal evolution (model *BM*; movie frame rate $f = 133.3$ frames s$^{-1}$) is available https://youtu.be/w8ZJqZK63KY. For comparison, a short movie showing the explosion in the diffuse ISM (model *BL*; $f = 133.3$ frames s$^{-1}$) or the temporal evolution without SN explosion (model *BO*; $f = 33.3$ frames s$^{-1}$) is available https://youtu.be/byw8LQ38i8M and https://youtu.be/36nB1aAL-2o, respectively. The movies and data are publicly available at https://etsin.fairdata.fi/dataset/602bb9a6-0626-43db-9073-054bd3332fff/data.

## Discussion

We have conducted high-resolution MHD simulations that explicitly follow dust destruction by the combined effects of grain-grain collisions and sputtering of an SN blast wave in a turbulent multiphase, magnetized ISM. Several factors affect dust processing induced by an SN, but also the background processing due to ISM dynamics is considerable (see green profiles in Fig. 5). The background processing rate outside the SN shock front is 11 M$_\odot$ Myr$^{-1}$ which is equivalent to 650 M$_\odot$ Myr$^{-1}$ kpc$^{-3}$. The ISM density variability created by turbulence can enhance the dust processing rate (see e.g., refs. 37–40), but an inhomogeneous ISM also appears to protect dust from propagating SN shocks.

In regions of moderate mean ambient density $\langle n_{\rm gas} \rangle \approx 1\,{\rm cm}^{-3}$, a considerable amount of dust near the explosion epicentre is lost early in the blast wave. Stars associated with OB clusters may explode in regions of higher ambient densities ($\gg 1\,{\rm cm}^{-3}$). How commonly this occurs, or whether they evacuate the ambient medium in which subsequent explosions occur, is unclear. Only 15% of SNe surveyed[41] interact with the small fractional volume of the ISM that comprises high-density molecular clouds of order $10^5\,{\rm cm}^{-3}$. Such observational signatures of interaction with dense gas can even arise later in the life of a remnant. On the other hand, such emissions from cloud densities below $10^5\,{\rm cm}^{-3}$ are difficult to detect, so the SN rate in dense regions may be underestimated[42]. Concluding the separation of clouds from their stellar progeny occurs mainly ahead of the SNe[43], find it likely that SNe will more often occur in ambient diffuse ISM, subject to their limited resolution and absent ionization. To examine processing in much higher density locations, or earlier in remnant evolution would require resources and inclusion of additional physics beyond the scope of this paper.

Nevertheless, let us consider what to expect of dust destruction at high gas densities. Contrary trends are subject to the actual density, its spatial extent, and the shock velocity. If the ambient mass is sufficiently high the shock will quickly dissipate before it can penetrate the entire region, resulting in almost no dust processing in the outer regions. Therefore, the dust survival rate in the high-density region could be arbitrarily large if the region is sufficiently massive. On the other hand, where the mass is insufficient the shock would overrun the entire region and process all the material. The expected outcome still depends on the extent to which self-shielding effects due to higher densities compensate for the higher destruction due to greater dust abundances. In areas that are closest to the explosion centre, the destroyed dust mass will be increased due to higher sputtering and grain-grain collision rates, which is confirmed by the early dust evolution at moderate densities in this study and also by simulations of ejecta clumps that are overrun by shocks (e.g. ref. 36). In addition to adiabatic effects, radiative gas cooling is accelerated in high-density regions and consumes a substantial quantity of the energy available to

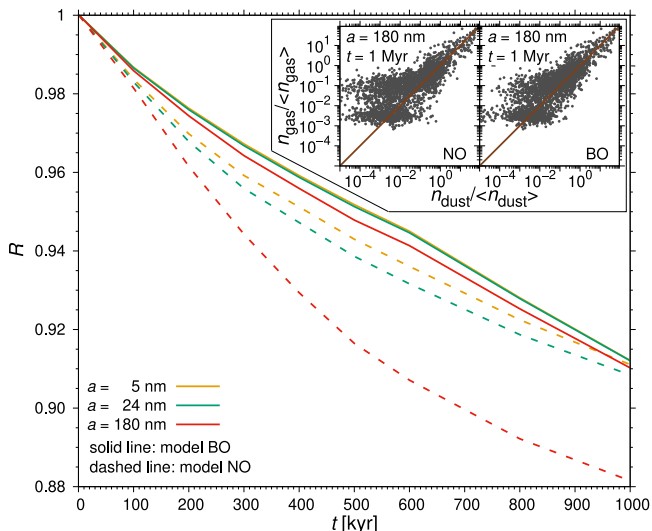

**Fig. 4 | Reduction of correlation between dust and gas.** Variation over time of the Pearson correlation coefficient $R$ due to the background dust processing between the logarithmic gas density and the logarithmic dust density for models *BO* (solid lines) and *NO* (dashed lines) at different grain sizes (different colours). The inset shows the scatter plots of normalized gas and dust density for the grain size 180 nm at 1 Myr, with the brown line indicating an exact correlation.

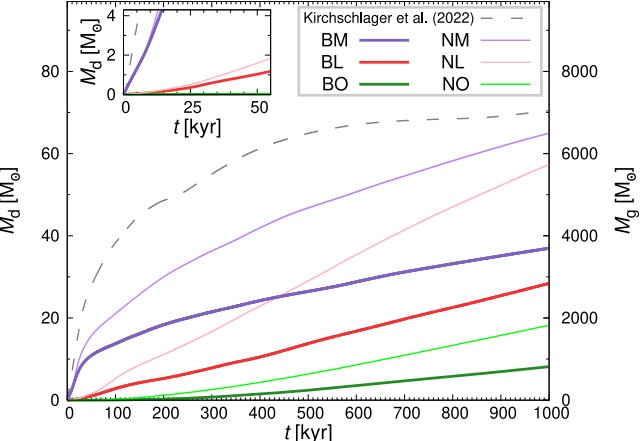

**Fig. 5 | Cumulative dust mass destroyed and cleared gas mass.** Total dust mass destroyed $M_d$ and cleared gas mass $M_g$ as a function of time, for each model as listed in the legend. The scales on each $y$-axis are directly equivalent, with $M_g = 100 M_d$. The inset shows the dust mass destroyed within the first 50 kyr.

the shocked gas, which in turn is no longer available for dust destruction processes. In summary, the balance between these conflicting effects is difficult to estimate and we shall require simulations to include sufficiently high-density regions.

The plausibility of the destroyed dust masses derived in ref. 14 has been checked (Section 5.1 in ref. 14) against the results of previous studies[10,27,32]. Deviations can be mainly retraced to different conditions (explosion energies, magnetic field strengths, gas-to-dust-mass ratios), neglecting physical processes (grain–grain collisions, kinetic sputtering), or different evolution times. In the present study, the higher density inhomogeneity induced by SN-driven turbulence reduces dust losses by 6–14 $M_\odot$ over 1 Myr relative to the uniform ambient ISM model of ref. 14. However, it is perhaps more indicative to consider the results over only 500 kyr, within which timescale neighbouring SNe might be expected to interact with the remnant. Thus far the multiphase inhomogeneous ISM reduces the dust destruction compared to ref. 14 by 18–36 $M_\odot$.

While the authors anticipated that including the effects of the Lorentz force might affect dust destruction, its impact is surprisingly strong. When including the Lorentz force acting on charged grains within 1 Myr about 28–29 $M_\odot$ of dust is additionally conserved in the multiphase ISM than without Lorentz forces. Within 500 kyr the total destroyed dust mass is 13.8 (26.5) $M_\odot$ for the low (moderate) density explosion site. Lorentz forces on charged dust reduce destructive grain–grain interaction (in particular fragmentation of large grains), which lowers the overall dust-destruction rate. Fewer small grains are produced. The magnetic field at least halves the dust losses due to background processing, but has even more impact against SN shocks. The lower dust destruction in the ISM at higher magnetic fields also confirms the results of previous studies (e.g., ref. 10).

Due to the absence of both galactocentric differential rotation and stratification, here large-scale dynamo is not present. The magnetic field generated purely by a small-scale dynamo has only a turbulent structure and saturates at a strength an order of magnitude weaker than might be expected in disk galaxies[31,44–46]. The large-scale dynamo adds a strong field ordered along the plane of the disc[47,48] but also entrains a turbulent field over ten times stronger than obtained here[49]. It is likely that this stronger turbulent component would protect dust even more effectively. Large-scale fields are observed to be

weaker than the turbulent component[50], so are unlikely to undermine, and may even enhance, the effect of the turbulent field.

In these models, we assume an initial dust abundance proportional to the gas density (initial gas-to-dust-mass ratio 100). It is not our aim to study the early Universe or pristine ISM, so a much higher gas-to-dust-mass ratio can be ignored here. We find from all models for regions with only background processing that the dust tends to aggregate in more filamentary structures than the gas. A turbulent magnetic field appears to cluster the dust on smaller scales and with reduced filamentary structure. Charged grains seem to be better protected from destruction in the presence of a magnetic field. We hypothesize that this is due to the fact that the Larmor time is affected by the variation of the local Alfvénic Mach number, which means the magnetic force on the dust alters dust dynamics and hence the clustering of dust grains.

Dust survival in the ISM as an SN shock wave propagates through it can vary due to mainly three factors: the distribution (inhomogeneity) of the ISM gas, the mean density of the gas, and the ambient dust abundances at the site of the SN. The importance of the latter two factors was quite expected. The first has a surprisingly strong effect. When factoring in the effect of the Lorentz force from a turbulent magnetic field these have a net result stronger than we anticipated. Overall, the three factors listed above reduce dust losses compared to the homogeneous case of the reference model[14] without the Lorentz force by between 28% and 55% within 500 kyr. Including the effects of the turbulent magnetic field reduces this even further, overall by between 60% and 79% (see Table 1).

A destroyed dust mass of around 30 $M_\odot$ (including turbulence and magnetic fields) is higher than found in other studies, e.g. in ref. 10 (about 10 $M_\odot$). However, we can reconcile the deviation by accounting for the larger explosion energy ($1 \times 10^{51}$ ergs vs. $0.5 \times 10^{51}$ ergs), lower gas-to-dust-mass ratio (100 vs. 163), and longer evolution times (1 Myr vs. 540 kyr). Although including turbulence and magnetic fields reduces the destroyed dust mass by a factor of about 2, the total amount of destroyed dust is still very high. For conditions as in ref. 10 this would result in destroyed dust masses of ~5 $M_\odot$. Assuming that a single core-collapse SN produces dust of order 1 solar mass, it is significantly less than the dust destruction, and thus a net dust destroyer. Including magnetic fields and turbulence significantly reduces the burden for dust production sources, namely AGB stars (not in the early Universe) or the ISM[51,52], necessary to account for levels of net dust in the universe. Further effects that could potentially reduce the dust

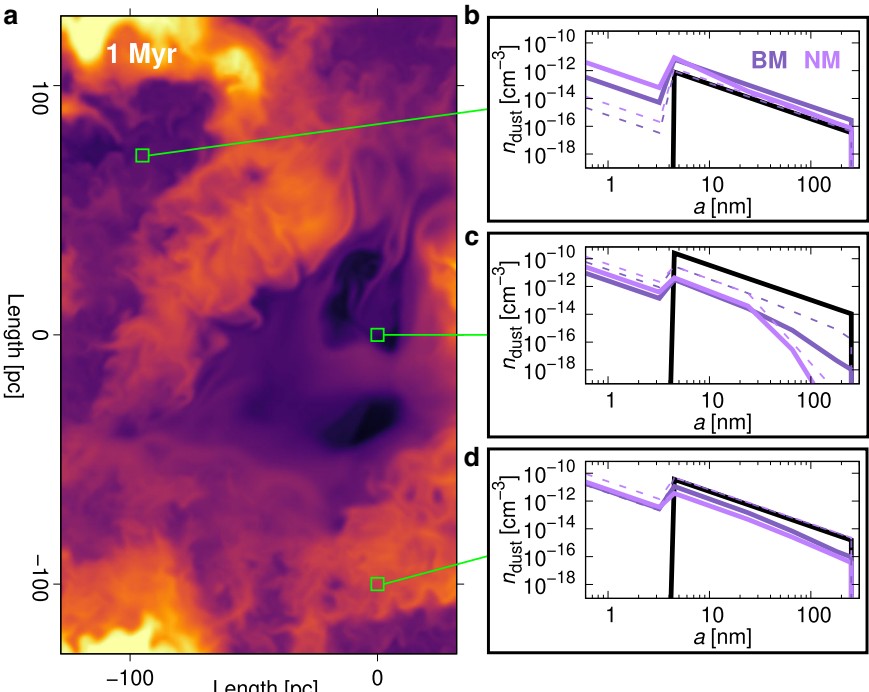

**Fig. 6 | Regional variation of dust distribution at various times. a** Location within the domain with an explosion in the moderate ambient gas density of three specimen regions (green boxes) representing two unshocked regions (top and bottom) and the shocked region (centre). The green boxes have a size of 5 pc × 5 pc. **b** The dust density initial MRN distribution (black), distribution after 200 kyr (dashed lines), and after 1 Myr (solid lines) for an unshocked region of low gas density, with Lorentz force (dark-purple; model *BM*) and without Lorentz force (light-purple; model *NM*). **c** Dust density distribution for the central shocked region. **d** Dust density distribution for an unshocked region of moderate gas density. The dust distributions show the average density for each grain size in each specimen. In the shocked region, a significant amount of dust grains is removed or destroyed by the shock while the change in the unshocked regions is due to background processing (transport and destruction).

destruction, at least in the first few kyr, e.g. stellar wind-blown bubbles and shells[27] or large-scale dynamos, are promising and have to be considered in future studies.

## Methods
### Numerical simulations
For the ambient turbulent ISM, within which we process the dust, as an initial condition we use a snapshot at an MHD statistical steady state

**Table 2 | The cooling function parameters Eq. (6)**

| $T_k$ [K] | $\Lambda_k$ [ erg g$^{-2}$ s$^{-1}$ cm$^3$ K$^{-\beta_k}$ ] | $\beta_k$ |
|---|---|---|
| 0 | 0 | ... |
| 90 | 3.70e16 | 2.12 |
| 141 | 9.46e18 | 1.00 |
| 313 | 1.18e20 | 0.56 |
| 6102 | 1.10e10 | 3.21 |
| 1e5 | 1.24e27 | −0.20 |
| 2.88e5 | 2.39e42 | −3.00 |
| 4.73e5 | 4.00e26 | −0.22 |
| 2.11e6 | 1.53e44 | −3.00 |
| 3.98e6 | 1.61e22 | 0.33 |
| 2.00e7 | 9.23e20 | 0.50 |
| ∞ | ... | ... |

The cooling coefficient $\Lambda_k$ applies to $T^{\beta_k}$ for temperature $T_k \leq T < T_k$.

from a three-dimensional simulation of supernova-driven turbulence as reported in ref. 46. In a periodic Cartesian domain of 256 parsecs along each dimension with a mean gas number density of 1 cm$^{-3}$, a weak random magnetic field is amplified through dynamo to a mean energy density of about 5% equipartition with the mean kinetic energy density. The model has a grid resolution size of 0.5 parsecs along each edge.

We do not resolve self-gravity. The maximal cold gas number density minimally of 600 K is a few tens cm$^{-3}$ so that the corresponding Jeans length $\lambda_J$ of 43 parsecs exceeds the size of such structures in the model. Taking the mean density of 1 cm$^{-3}$ and a mean sound speed of 12 km s$^{-1}$ in the warm gas, $\lambda_J >> 256$ pc. Effects of self-gravity on dust cannot be entirely ruled out[53], but we assess that these effects are small in the present simulations, in particular in the cases that include the Lorentz force. The maximal gas densities we can resolve are constrained by the limited practicable resolution required to span a domain size of 256 pc and adequately capture multiple SN remnants. The resulting turbulence must be evolved tens of Myr to saturate a small-scale dynamo. Increased resolution, with increased maximal densities and reduced cooling times, increases by an order of magnitude both the size of each numerical integration and the total integration time. Resolving molecular clouds is therefore beyond the scope of this study. Similarly, cooling by gas−dust interaction is not taken into account because it would require that dust physics is treated within the MHD simulation, at a huge computational expense, while the impact on the thermal sputtering rate is modest.

The turbulence is driven by SNe distributed uniformly randomly in space at a Poisson rate of approximately 1 Myr$^{-1}$ within the domain.

Equivalent to a rate of around 20% of that of the Solar neighbourhood, this lower rate maintains a multiphase ISM with appropriate fractional volumes of cold, warm and hot gas. In the periodic box, with no escape for hot gas a higher SN rate would quickly induce thermal runaway[54], leaving the computational domain saturated by hot gas and a tiny fraction of very dense cold and warm gas. Disk stratification with a halo into which hot gas can escape, cool and recirculate is omitted.

At the time at which we apply the dust-processing model to this simulation, we cease the continuous random SN explosions, in order to isolate the effects during the lifespan of a single SN remnant. Using the same ambient state we consider the case of two SN explosions, one in a diffuse region of the ISM, with $n$ -0.03 cm$^{-3}$, and another in a more dense region, with $n$ -0.7 cm$^{-3}$. We also run a control model, in which no new SN is added to the ambient state to isolate the dust processing induced by the turbulent background dynamics from that of the SN blasts.

To model the explosion we inject $10^{51}$ erg of thermal energy with a spherical Gaussian profile of radial scale 8 pc. At this resolution and ambient gas density, it is not necessary to include momentum injection[55,56] to obtain sufficient kinetic energy. In the highly dynamic and inhomogeneous ISM, the ideal analytic solutions of Sedov–Taylor do not apply but can be used[57] to verify the model rapidly evolves to match the adiabatic solution within a few thousand years, and well before subsequently reproducing the snowplough evolution. We omit SN mass ejecta, as the SN shockwave travels rapidly beyond the extent of the ejecta. The snowplough phase and the interaction with the surrounding ISM has essentially no connection with the properties of the ejecta, which we can thus safely ignore. The ambient density and flow are not altered to inject the SN, so as the flow evolves from the thermal pressure, it immediately interacts with the turbulent interior gas and magnetic field. Inevitably, in the first few thousand years dust destruction will be understated, but this would be true for all models, including with the uniform ambient medium. Thus, a comparison between our models provides a reliable indication of the relative effects explored.

We further omit an evacuated bubble around the progenitor star created by radiation pressure or stellar winds prior to the SN explosion. Cleared of most of the gas and dust, these bubbles and the surrounding wind-driven shells extend in a homogeneous medium with 1 particle per cm$^{-3}$ up to 25 pc after 1 Myr exposure time[27]. The blast wave needs only a few kyr to reach these distances. The influence of these bubbles would affect only a small fraction of our model domain and only the first < 1% of the blast wave evolution time. Though the blast wave velocity and strength can be disturbed beyond that, we expect a low impact of an evacuated bubble on the total mass of destroyed dust.

For the generation of the multiphase MHD turbulence used for our ambient ISM and its further evolution with or without our isolated SN explosion, we solve the set of nonideal compressible MHD equations, using the sixth-order Pencil Code[58] PDE solver. As presented in[14] we include radiative cooling and UV heating processes, which apply the rate of cooling depending only on a piece-wise varying exponent of temperature and normalized by gas density. Cooling approximates the cumulative processes applying at metallicity abundances expected in the Solar neighbourhood[59]. This neglects the impact the dust could have on the thermodynamics of the gas if the dust together with a more sophisticated cooling dependent on the dust abundances were included in the MHD model. It is reasonable to hypothesize, from our results regarding dust-gas decoupling, that such an effect would even further reduce dust destruction, as cooling would be even more effective where dust density is highest.

In this study, following[31], we solve the set of nonideal MHD equations

$$\frac{D\rho}{Dt} = -\rho\nabla\cdot\mathbf{u} + \nabla\cdot\zeta_D\nabla\rho, \tag{1}$$

$$\begin{aligned}\rho\frac{D\mathbf{u}}{Dt} = &-\rho c_s^2\nabla\left(s/c_p + \ln\rho\right) + \mu_0^{-1}\nabla\times\mathbf{B}\times\mathbf{B} \\ &+ \nabla\cdot(2\rho\nu\mathbf{W}) + \rho\nabla(\zeta_\nu\nabla\cdot\mathbf{u}) \\ &+ \nabla\cdot\left(2\rho\nu_6\mathbf{W}^{(5)}\right) - \mathbf{u}\nabla\cdot(\zeta_D\nabla\rho),\end{aligned} \tag{2}$$

$$\frac{\partial\mathbf{A}}{\partial t} = \mathbf{u}\times\mathbf{B} + \eta\nabla^2\mathbf{A} + \eta_6\nabla^6\mathbf{A}, \tag{3}$$

$$\begin{aligned}\rho T\frac{Ds}{Dt} = &E_{th}\dot{\sigma}h^{-1} + \rho\Gamma - \rho^2\Lambda + \eta\mu_0^{-1}|\nabla\times\mathbf{B}|^2 \\ &+ 2\rho\nu|\mathbf{W}|^2 + \rho\zeta_\nu(\nabla\cdot\mathbf{u})^2 \\ &+ \nabla\cdot\left(\zeta_\chi\rho T\nabla s\right) + \rho T\chi_6\nabla^6 s \\ &- c_\nu T\left(\zeta_D\nabla^2\rho + \nabla\zeta_D\cdot\nabla\rho\right),\end{aligned} \tag{4}$$

and includes the ideal gas equation of state $i$, for which the adiabatic index is 5/3. Treating the ISM as a monatomic, fully ionized plasma we apply a mean molecular weight of 0.531. Common variables and symbols take their usual meanings. $s$ is specific entropy and $W$ is the traceless rate of the strain tensor, with $W^{(5)}$ its fifth-order application to hyperdiffusion. Viscosity $\nu = 5\times10^{-4}$ kpc km s$^{-1}$ and magnetic diffusivity $\eta = 10^{-4}$ kpc km s$^{-1}$. Shocks are resolved with artificial viscosities $\zeta_D$, $\zeta_\nu$ and $\zeta_\chi\propto\nabla\cdot u$, only where flows are convergent. Sixth-order hyperdiffusion applies coefficients $\nu_6 = \eta_6 = \chi_6 = 6.25\times10^{-16}$ kpc$^5$ km s$^{-1}$. $\nabla^6 = \partial_i^3\partial_i^3$.

$E_{th} = 10^{51}$ erg of SN energy are injected in Eq. (4) at a Poisson rate of about 60 kpc$^{-3}$ Myr$^{-1}$. Background ultraviolet heating is applied in Eq. (4) as

$$\Gamma = \frac{\Gamma_0}{2}\left(1 + \tanh\left[\frac{2\cdot10^4\,\mathrm{K} - T}{2000\,\mathrm{K}}\right]\right), \tag{5}$$

with $\Gamma_0 = 0.0147$ erg g$^{-1}$ s$^{-1}$. The radiative losses $\Lambda$ in Eq. (4) are modelled via a piece-wise power law dependence on the temperature of the form

$$\Lambda(T) = \Lambda_k T^{\beta_k} \quad \text{for} \quad T\in[T_k, T_{k+1}), \tag{6}$$

with the parameters as listed in Table 2.

For the initial state of the dynamo simulation uniform, ISM has gas number density $n = 1$ cm$^{-3}$ and temperature $T = 10^4$ K. The seed field comprises white noise with a mean strength of 1 nG. A snapshot from the saturated state of the dynamo is used as the initial condition for all of the six cases explored in this study.

We use the same MHD velocity field and gas structure to model the dust processing with and without the Lorentz force arising from its magnetic field for each case with an SN explosion and without.

## Dust processing methods

The evolution of the dust grains is driven by the gas conditions and by the magnetic field in the turbulent, inhomogeneous and shocked ISM. Pencil provides snapshots at intervals of 250 yr from the 3D gas density, gas velocity, gas temperature and magnetic field. We use our post-processing code PAPERBOATS[36] to study the transport, destruction and gas accretion of the dust grains for these snapshots. To follow the dust evolution, the 'dusty-grid approach' is used where

the dust location is discretized to spatial cells and the dust in each cell is apportioned in different grain size bins. The dust grains can move both spatially as well as between the grain size bins as a result of dust destruction or growth during a time-step.

Due to the excessive computational effort required for highly resolved 3D post-processing simulations, we confine the study of dust-processing to a thin slice through the centre of the explosion, spanning a cuboid volume of 256 pc × 256 pc × 0.5 pc. The destroyed dust masses are computed for this thin slice and multiplied by 512 in order to estimate the destroyed dust masses of the entire 3D domain, under the assumption that the dust processing is approximately isotropic. For the simulations including the blast wave the destroyed dust masses are scaled by a correction factor of 0.365, which takes into account that the explosion takes place in a central spherical region and that the blast wave does not reach the outer edge of the domain within 1 Myr. The value of 0.365 was derived by comparing the destroyed dust masses in the thin slice of the homogeneous ambient ISM (as in ref. 14) to the inhomogeneous ISM, and expecting that the same ratio exists for the 3D domain between the homogeneous and the inhomogeneous ISM. We recognize that the remnant is far from spherical and the turbulence far from isotropic in this system. The slice is an arbitrary selection and it is reasonable to assume that it is typical of the system. When resources permit a stochastic study across multiple slices and realizations of the ISM would better constrain these assumptions.

In this study, the dust transport, dust destruction and dust growth are determined. The acceleration of dust grains of mass $m$ occurs by gas–grain collisions, by Coulomb interactions of charged grains in the ionized gas as well as by Lorentz forces on charged grains in magnetic fields[60]

$$\mathbf{a}_{acc} = \frac{\mathbf{F}_{drag}}{m} + \frac{\mathbf{F}_{Lorentz}}{m}. \tag{7}$$

The net drag force caused by collisional drag and by plasma drag[2,61] is given as

$$F_{drag} = 2\sqrt{\pi} k_B T_{gas} a^2 \sum_j n_{gas,j} \left( \mathcal{F}_{col,j} + \mathcal{F}_{pla,j} \right), \tag{8}$$

where $k_B$ is the Boltzmann constant, $T_{gas}$ is the gas temperature, $a$ is the grain size, and $\mathcal{F}_{col,j}$ and $\mathcal{F}_{pla,j}$ are the 'Collisional term' and the 'Plasma term', respectively. The sum runs over all plasma species $j$ within the gas (atoms, molecules, ions, and electrons). The Lorentz force is given by

$$\mathbf{F}_{Lorentz} = Q_{grain} \mathbf{v}_{rel} \times \mathbf{B}, \tag{9}$$

where $Q_{grain}$ is the dust grain charge, $\mathbf{B}$ is the magnetic field, and $\mathbf{v}_{rel}$ is the relative velocity between dust grain and magnetic field. Dust grain charges are determined due to the ionization of impinging plasma particles (ions and electrons), associated secondary electrons, transmitted plasma particles, and field emission[62].

The dust material can be destroyed by either sputtering or grain–grain collisions. Sputtering[1,2,63] is the ejection of grain atoms due to the bombardment of gas particles (atoms, ions, or molecules). We distinguish between kinetic sputtering (velocity of gas particles due to the grain moving relatively to the gas) and thermal sputtering (thermal motion of the gas particles). The rate of decrease of grain radius $a$ per unit time, $da/dt$, due to kinematic sputtering and thermal sputtering can be expressed as

$$\frac{da}{dt} = \frac{\langle M_{atom} \rangle}{2\rho_{bulk}} v_{rel} \sum_k n_{gas,k} Y_k(E) \tag{10}$$

and

$$\frac{da}{dt} = \frac{\langle M_{atom} \rangle}{2\rho_{bulk}} \sum_k n_{gas,k} \langle Y_k v \rangle, \tag{11}$$

respectively, where $\langle M_{atom} \rangle$ is the average atomic mass of the grain atoms, $\rho_{bulk}$ is the material density, $v_{rel}$ is the relative velocity between dust grains and the surrounding gas, $n_{gas,k}$ is the number density of gas species $k$, $Y_k(E)$ is the sputtering yield of gas species $k$ as a function of the kinetic energy $E$, $\langle Y_k v \rangle$ is the sputtering yield averaged over the Maxwellian velocity distribution, and $v$ is the thermal velocity of a gas particle. For temperatures higher than $10^4$ K the relative velocity between grain and the surrounding gas is not unimodal but is a combination of the motion of the grain relative to the surrounding gas and the thermal motion of the gas particles. These two motions are combined using a skewed Maxwellian distribution instead of a regular Maxwellian distribution[9,64]. The size-dependent sputtering effect is also taken into account[65].

The second important kind of dust destruction process is grain–grain collisions. Collisions between dust grains of different sizes occur due to the relative velocities between them. The collision probability[36] of a single grain of size $a_i$ to collide with any grain of size $a_j$ during the time interval $\Delta t$ is

$$P_{ij} = 1 - \exp \left[ - n_j \sigma_{col} v_{col} \Delta t \right], \tag{12}$$

where $n_j$ is the number density of grains of size $a_j$, $\sigma_{col}$ is the collision cross-section, and $v_{col}$ is the collision velocity. Both $\sigma_{col}$ and $v_{col}$ take repulsion or attraction due to Coulomb interaction between the charged dust grains into account. The outcome of a grain–grain collision[6,13] depends on the collision energy. For the largest collision velocities (>19 km/s), the dust grains can be fully vaporized which means that the whole material goes into the gas phase. Please note that a partial vaporization approach as in ref. 60 is not included but has shown to be mainly important for sizes larger than the MRN[35] grains. On the other hand, intermediate collision velocities result in (partial) shattering of the dust grains[37]. The shattered material is then redistributed in a size distribution of fragments. At collision velocities below 2.7 km/s, the grains are not shattered but bounce or even stick together, though the latter process has a low occurrence in the simulations as the gas and dust velocities are too high. Further dust growth processes like gas accretion or ion trapping[66] of destroyed dust material also play only a minor role. We neglect gas accretion and ion trapping of the regular gas due to the nature of post-processing[66].

Dust processing with and without the Lorentz force acting on the dust is considered to assess its effect on the dust evolution. In either case, the gas velocities and structure are obtained from the same MHD simulation. The initial spatial distribution of the dust follows the gas, assuming a constant gas-to-dust-mass ratio of 100. The grains are made of silicate and follow initially a size distribution of MRN type. The grains are binned in 20-size bins with an additional collector bin at the lower and upper end of the size distribution, respectively.

It is easier to isolate the location of SN explosions modelled in a uniform ambient ISM in order to calculate the dust processing associated with the remnant. In such an inhomogenous ambient ISM as applies here, this is difficult to do. We, therefore, include a model without an SN explosion, so we can subtract the dust processing that occurs absent the SN, background processing. While the MHD model has a mean gas number density of 1 cm$^{-3}$, due to its turbulent evolution the initial mean density of the two-dimensional slice on which we model the dust processing is 0.59 cm$^{-3}$. For the duration of the dust processing models, the mean gas density has negligible monotonic decay (Supplementary Fig. 7). but locally can fluctuate modestly due to the advection of gas in and out of the plane within the three-dimensional turbulence.

Our results of purely background processing show that a statistical steady state filamentary dust distribution evolves from the initial state within about 300 kyr. Dust processing in the early aftermath of an explosion in future simulations could, therefore, be improved by evolving the dust processing 300 kyr before exploding the SN.

## Data availability

The Pencil Code simulation run files and the initial snapshot used at the beginning of the dust processing experiment (about 20 GB) have been deposited in the Finnish Fairdata storage service under the accession code https://doi.org/10.23729/ac4542ad-ab85-4ccc-8d0c-60bfc5472ef2 from which the MHD solutions can be reproduced without having to replicate the dynamo simulations. The start and run files are also included for replicating the dynamo simulations. The time series of 2D slices alone of gas density, temperature, velocity and magnetic field, and the subsequent slices of dust densities used in the analysis exceed 1.4 TB. Public hosting is impractical for the authors due to this large size. However, the datasets generated and analysed during the current study are available from the corresponding author on request.

## Code availability

We use the Pencil Code[58] to perform all simulations, which is freely available under https://github.com/pencil-code/https://doi.org/10.21105/joss.02807. The Paperboats code[14,36,60,66] is available at https://doi.org/10.5281/zenodo.10036806.

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

## Acknowledgements
We acknowledge funding from the European Research Council (ERC) under the European Union's Horizon 2020 research and innovation programmes: SNDUST ERC-2015-AdG-694520 (F.K.), DustOrigin ERC-2019-StG-851622 (F.K.) and UniSDyn grant no. 818665 (F.A.G.); the Swedish Research Council (Vetenskapsrådet), grants no. 2015-04505 (L.M.) and 2022-03767 (L.M.); the Academy of Finland ReSoLVE Centre of Excellence grant 307411 (F.A.G.); and the Ministry of Education and Culture Global Programme USA Pilot 9758121 (F.A.G.). We appreciate the generous computational resources from CSC-IT Centre for Science, Finland, under Grand Challenge GDYNS Project 2001062.

## Author contributions
F.K., L.M. and F.A.G. have made essential contributions to the conceptualization, data analysis and writing of this paper.

## Funding

## Competing interests
The authors declare no competing interests.
