## [Peer Review File · Nature Communications]

Editorial Note: This manuscript has been previously reviewed at another journal that is not operating a transparent peer review scheme. This document only contains reviewer comments and rebuttal letters for versions considered at *Nature Communications*.REVIEWER COMMENTS

Reviewer #1 (Remarks to the Author):

As with my report on the previous version of this manuscript for Nature Astronomy, I have no major concerns about the numerical methods employed in this paper, which clearly represent the state-of-the-art in modelling dust destruction by SNe. I agree that the reduction in the mass of dust destroyed found by the authors, caused by the inclusion of turbulence and magnetic fields, is an interesting result. However, I still think the authors are not accurately representing the impact and importance of their results to the field as a whole.

The single sentence in the abstract that I took issue with is quite an important one, providing the key motivation for the present study. The current version states 'Theory and prior simulations of dust processing by supernovae in uniform ISM predict much greater dust destruction than is compatible with estimates of dust abundances observed in galaxies.' This is simply untrue. It might arguably be the case for a few extreme high-redshift objects, as the authors mention in section 1, but the abstract, as written, gives the misleading impression that all observed dust masses in galaxies are incompatible with the previously-estimated destruction efficiencies. The authors should either replace this sentence with something more accurate, or specify that this incompatibility is under the highly-implausible assumption that CCSNe are the only sources of dust.

Another point that the authors have not really addressed is that their 30 Msun value for the mass of dust destroyed per SN, including turbulence and magnetic fields, is significantly higher than the widely-accepted value of ~ 10 Msun from the previous literature (e.g. Slavin+15). The statement in section 3 that these results represent a 'confirmation of relatively low net destruction rates' is also untrue; if we take the authors' results at face value, the dust budget crisis has gotten worse, rather than the gap between SN destruction and production closing. Even applying the same reduction factor of 80% to the lower 10 Msun value, SNe are still net destroyers of dust. While the results on the impact of the magnetic field are interesting, they do not significantly change our understanding of dust evolution in the ISM, as some additional non-SN source of dust is still necessary (and presumably dominant). The factor of ~ 8 difference between the authors' paper I and the prior literature, compared to a factor of 3-5 difference from including magnetic fields and turbulence, also suggests that these additional effects are of comparable or lesser importance to modest variations in parameters such as the explosion energy or dust-to-gas ratio. None of these points are adequately addressed in the discussion.

Reviewer #2 (Remarks to the Author):

The authors have explicitly acknowledged the computational challenges they would face in modeling the evolution of SNRs in significantly higher density media. While I understand the task is very challenging and resource-consuming, I insist they should at least hint at what to expect at higher gas densities given the reduced cooling times. Nevertheless, I believe the manuscript is in good shape and merits publication, as it lays a foundation for further discussion in the literature, allowing others to join the general discourse on the topic.

Recommendation to the Authors:

Regarding the statement "Hewitt & Yusef-Zadeh (2009) determine that only 15% of SNe surveyed occur in the small fractional volume of the ISM that comprise high density molecular clouds" it is crucial to provide additional context and clarification. The 15% figure specifically pertains to the fraction of maser-emitting supernova remnants (SNRs) within the high-density ($\sim 10^5 \text{ cm}^{-3}$) galactic molecular ring, highlighting the presence of masers in a subset of SNRs within this region.

However, it is pertinent to consider the existence of a hidden population of rapidly evolving supernova remnants within high-density molecular clouds that quickly become radio-quiet (Sofue 2020, PASJ 72L, 11S). These radio-quiet 'buried' SNRs could represent an important fraction of supernova events that are challenging to detect through traditional methods.

I recommend revising the statement to accurately represent Hewitt et al.'s findings regarding maser-emitting SNRs, while also acknowledging the importance of further investigating the presence and significance of SNRs in high-density molecular clouds. Regarding the study of Gatto et al. (2015), they acknowledged similar limitations as the present study; i.e. limited resolution to address the impact of SNe in high-density regions, and the need to include stellar clustering, stellar winds, and ionizing radiation.

ÿbResponse to the referees, September 2023

We appreciate the referee s report on the manuscript entitled "Supernova dust destruction in the magnetized turbulent ISM", ref. NCOMMS-23-29020-T, by Florian Kirchschrager, Lars Mattsson and Frederick A. Gent, which was submitted to Nature Communications.

According to the referee s comments we have conducted a revision of this manuscript, and we hope the clarity of the paper has improved. Revisions from the previous manuscript are shown in red font in the revised manuscript.

Kind regards,

Florian Kirchschrager, Lars Mattsson and Frederick Gent

COMMENTS Reviewer #1:

1) As with my report on the previous version of this manuscript for Nature Astronomy, I have no major concerns about the numerical methods employed in this paper, which clearly represent the state-of-the-art in modelling dust destruction by SNe. I agree that the reduction in the mass of dust destroyed found by the authors, caused by the inclusion of turbulence and magnetic fields, is an interesting result. However, I still think the authors are not accurately representing the impact and importance of their results to the field as a whole.

The single sentence in the abstract that I took issue with is quite an important one, providing the key motivation for the present study. The current version states 'Theory and prior simulations of dust processing by supernovae in uniform ISM predict much greater dust destruction than is compatible with estimates of dust abundances observed in galaxies.' This is simply untrue. It might arguably be the case for a few extreme high-redshift objects, as the authors mention in section 1, but the abstract, as written, gives the misleading impression that all observed dust masses in galaxies are incompatible with the previously-estimated destruction efficiencies. The authors should either replace this sentence with something more accurate, or specify that this incompatibility is under the highly-implausible assumption that CCSNe are the only sources of dust.

The reviewer has made a clearer point above. We see the issue now. We have now completely rewritten the abstract. In particular, we rephrased the sentence mentioned by the referee.

Furthermore, we revised three sentences in the Introduction to avoid making the impression that all observed dust abundances in galaxies are incompatible with previously estimated destruction efficiencies. The point we want to make is that the overall destruction efficiency must be such that the dust masses of all types of galaxies can be explained.

2) Another point that the authors have not really addressed is that their 30 Msun value for the mass of dust destroyed per SN, including turbulence and magnetic fields, is significantly higher than the widely-accepted value of ~10 Msun from the previous literature (e.g. Slavin+15). The statement in section 3 that these results represent a 'confirmation of relatively low net destruction rates' is also untrue; if we take the authors' results at face value, the dust budget crisis has gotten worse, rather than the gap between SN destruction and production closing. Even applying the same reduction factor of 80% to the lower 10 Msun value, SNe are still net destroyers of dust. While the results on the impact of the magnetic field are interesting, they do not significantly change our understanding of dust evolution in the ISM, as some additional non-SN source of dust is still necessary (and presumably dominant). The factor of ~8 difference between the authors' paper I and the prior literature, compared to a factor of 3-5 difference from including magnetic fields and turbulence, also suggests that these additional effects are of comparable or lesser importance to modest variations in parameters such as the explosion energy or dust-to-gas ratio. None of these points are adequately addressed in the discussion.

We have completely revised the last paragraph of Section 3 (Discussion), addressing the points raised by the referee.

COMMENTS Reviewer #2:

1) The authors have explicitly acknowledged the computational challenges they would face in modeling the evolution of SNRs in significantly higher density media. While I understand the task is very challenging and resource-consuming, I insist they should at least hint at what to expect at higher gas densities given the reduced cooling times. Nevertheless, I believe the manuscript is in good shape and merits publication, as it lays a foundation for further discussion in the literature, allowing others to join the general discourse on the topic.

We have added a new paragraph in Section 3 (Discussion) in which we discuss

opposing effects for the dust destruction rates at high gas densities. Although concrete results are difficult to estimate, we give a rough outlook about dust survival in high density regions.

Recommendation to the Authors:

2) Regarding the statement "Hewitt & Yusef-Zadeh (2009) determine that only 15% of SNe surveyed occur in the small fractional volume of the ISM that comprise high density molecular clouds" it is crucial to provide additional context and clarification. The 15% figure specifically pertains to the fraction of maser-emitting supernova remnants (SNRs) within the high-density ($\sim 10^5 \text{ cm}^{-3}$) galactic molecular ring, highlighting the presence of masers in a subset of SNRs within this region.

We have clarified this interpretation as suggested.

3) However, it is pertinent to consider the existence of a hidden population of rapidly evolving supernova remnants within high-density molecular clouds that quickly become radio-quiet (Sofue 2020, PASJ 72L, 11S). These radio-quiet 'buried' SNRs could represent an important fraction of supernova events that are challenging to detect through traditional methods.

Agreed and added citation with context already included for comment 2.

4) I recommend revising the statement to accurately represent Hewitt et al.'s findings regarding maser-emitting SNRs, while also acknowledging the importance of further investigating the presence and significance of SNRs in high-density molecular clouds. Regarding the study of Gatto et al. (2015), they acknowledged similar limitations as the present study; i.e. limited resolution to address the impact of SNe in high-density regions, and the need to include stellar clustering, stellar winds, and ionizing radiation.

Added qualifying clause to use of citation.

We would add, that numerical simulations of SN driven turbulence de Avillez and Breitschwerdt, Gressel et al, Gent et al, Joung et al and Hill et al, find a resulting multiphase ISM structure which is too cold due to excessive gas cooling relative to observed warm and hot gas abundances, if the location of SNe are biased toward high densities, rather than random or clustered (which increases the proportion in diffuse locations). All of these are limited to resolution at best ~ 1 parsec, so the effect of cooling might be exaggerated.

REVIEWERS' COMMENTS

Reviewer #1 (Remarks to the Author):

I have no further comments on the paper, and I apologise to the authors if I did not make my point clearly in previous reports. I think the current version of the paper is suitable for publication.

Reviewer #2 (Remarks to the Author):

The authors have effectively addressed all of my previous concerns, and I am now ready to recommend the paper for publication.